# VALM: Variational Autoencoder Language Models for Highly Parallel Text Generation

## Abstract

Autoregressive language models have shown impressive abilities across domains. However, their token-by-token decoding limits inference speed. We introduce Variational Autoencoder Language Models (VALM), a non-autoregressive architecture that predicts entire sequences in parallel from a single global latent, with no denoising or diffusion losses. VALM uses a bidirectional transformer encoder and decoder with an ELBO objective, reducing sequential depth from $\mathcal{O}(LT)$ to $\mathcal{O}(L)$ for an $L$-layer network generating $T$ tokens. We train VALM-1, which generates 32 tokens in a single forward pass, demonstrating the applicability of pure VAEs to discrete text and presenting a novel approach to high-throughput language modeling on standard GPUs.

## 1 Introduction

Autoregressive language models (ARLMs) deliver strong results across NLP, math, and code, but inference is sequential: tokens are decoded one at a time. Latency scales with output length, and the sequential depth of an $L$-layer network is $\mathcal{O}(LT)$ for $T$ tokens.

Inspired by the success of diffusion in images, *diffusion LMs* have shown that ARLMs are not the only viable architecture for language modelling. They perform iterative denoising over $K$ steps in discrete token space or continuous embeddings (then discretize); quality improves with larger $K$, but latency scales with the number of iterations and hand-tuned schedules.

Before diffusion dominated vision, Variational AutoEncoders VAEs mapped a *single* latent to a full-resolution image in one shot: global structure is planned in the latent and rendered in parallel, so sequential depth depends on network depth, not output length. We port this idea to text: a single global latent $z$ conditions a bidirectional decoder that predicts all positions in parallel, collapsing sequential depth from $\mathcal{O}(LT)$ to $\mathcal{O}(L)$.

We propose VALM, a non-autoregressive variational language model that generates full sequences in a single forward pass. As shown in Figure 1, a bidirectional encoder compresses a target sequence into a global latent $z$ during training; a bidirectional decoder predicts all positions in parallel given $z$. Both components are standard Transformers with no causal masks. Training uses an evidence lower bound ELBO objective with a Gaussian prior $p(z) = \mathcal{N}(0, I)$; there is no denoising loss, no score matching, and no teacher distillation. At inference we sample $z \sim \mathcal{N}(0, I)$ once and decode all token logits at once.

**Contributions**

1. **Method.** VALM: a non-autoregressive VAE for text with bidirectional encoder and decoder that decodes all positions from a single global latent $z$ in one pass.
2. **Prototype.** VALM-1, a reference implementation that generates simple but coherent 32-token spans in one pass.
3. **Scaling signals.** Preliminary scaling on TinyStories and WikiText-103: loss improves predictably with parameters and data, without any collapse, suggesting room for further scaling.

Taken together, these results show that (i) single-pass language modeling is viable - AR or multi-step refinement is not the only option, (ii) a plain VAE without auxiliary AR or diffusion components

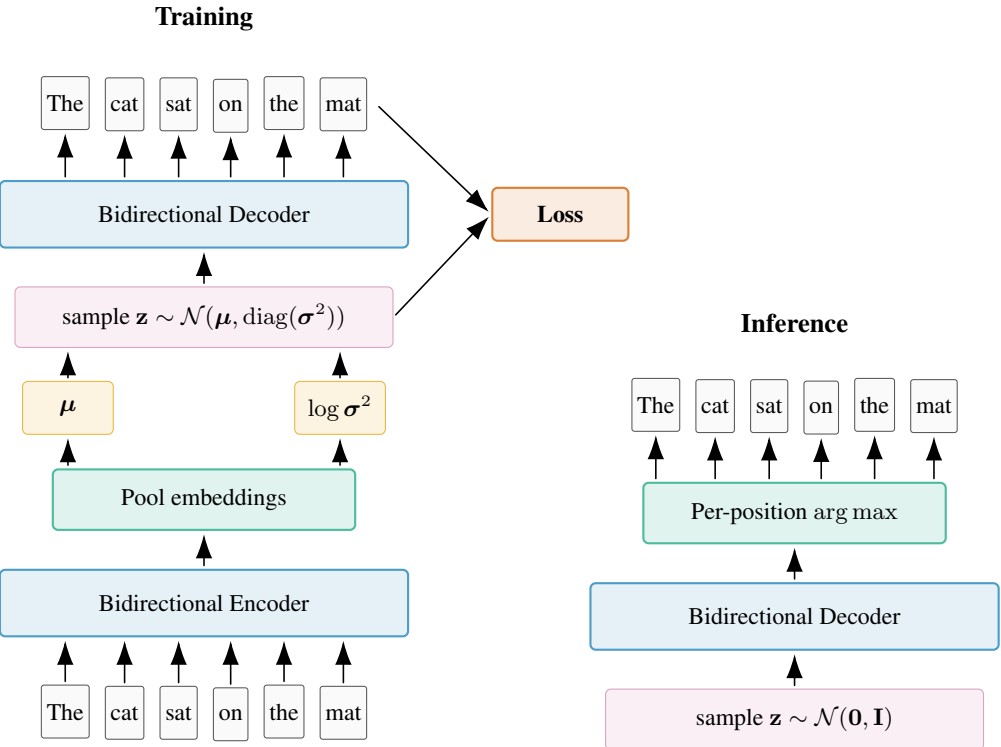

Figure 1: **VAE model architecture.** Left: Training. A bidirectional encoder pools and projects token embeddings to produce $\mu$ and $\log \sigma^2$; we sample $z$ via reparameterization and minimize masked token cross-entropy plus $\beta \, \mathrm{KL}\big(q_\phi(z \mid x) \, \| \, p(z)\big)$ (where KL denotes the Kullback-Leibler divergence). Right: Inference. Sample $z \sim \mathcal{N}(0, I)$; a bidirectional decoder predicts all positions in one forward pass given $z$ and absolute positions, then take per-position $\arg\max$. No autoregressive path or iterative refinement.

can model discrete text, and (iii) removing the AR path addresses the usual posterior-collapse issue in text VAEs because the decoder must use $z$.

## 2 METHOD

### 2.1 LANGUAGE MODELS

The goal of generative language modeling is to estimate a distribution over token sequences $x_{1:T} = (x_1, \ldots, x_T)$ drawn from a vocabulary $V$, i.e., $p_\theta(x_{1:T}) \in \Delta(V^T)$.

A standard approach is *autoregressive* (AR) modeling via the chain rule:

$$p_\theta(x_{1:T}) \; = \; \prod_{t=1}^{T} p_\theta\big(x_t \mid x_{<t}\big), \tag{1}$$

where each factor $p_\theta(\cdot \mid x_{<t})$ is a categorical distribution over $V$ with $\sum_{v \in V} p_\theta(x_t{=}v \mid x_{<t}) = 1$.

**Training.** Autoregressive LMs (ARLMs) are typically trained to minimize token-level cross-entropy:

$$\mathcal{L}_{\mathrm{AR}}(x) \; = \; -\sum_{t=1}^{T} \log p_\theta(x_t \mid x_{<t}). \tag{2}$$

**Inference cost.** Generation proceeds sequentially: $x_t \sim p_\theta(\cdot \mid x_{<t})$ (or $\arg\max$) for $t = 1, \ldots, T$. This imposes a token-by-token dependency: wall-clock latency scales as $\mathcal{O}(T)$, and the

sequential depth of an $L$-layer network scales as $\mathcal{O}(LT)$. Even with batched matrix multiplies, bandwidth and activation memory grow with sequence length $T$, limiting parallel speedups.[1]

## 2.2 WHY NAIVE PARALLEL DECODING FAILS

A naive way to remove the sequential bottleneck is to drop the conditionals from the AR factorization and assume tokenwise independence, predicting all positions in parallel,

$$\tilde{p}_\theta(x_{1:T}) = \prod_{t=1}^{T} \tilde{p}_\theta(x_t), \tag{3}$$

i.e., train a network that maps a constant or random input to per-position logits and minimizes cross-entropy.

This independence assumption is false for natural language: token choices depend on surrounding context, and the correct joint distribution cannot be reconstructed from per-token marginals. For example:

- **Collocations and names.** Independence makes *New Angeles/Los York* as likely as *New York/Los Angeles*.
- **Language consistency.** Trained on English and Chinese, an independent decoder freely mixes scripts within a sentence.
- **Sequence-level exclusivity.** On a toy corpus with only AAAA or BBBB (each w.p. $1/2$), the marginal optimum sets $P(x_t{=}\texttt{A}){=}1/2$ for all $t$, so the independent model generates mixes like ABAB, BBAB, etc., which have zero probability under the true data.
- **Word order and basic syntax.** Factorized models cannot enforce number agreement or canonical order (e.g., *the cat chased the mouse* vs. the ungrammatical *cat the mouse the chased*).

One might try to *manually* fix this by adding a small conditioning latent $z$ per sequence (e.g., a bit or scalar) encoding factors such as:

- collocation pattern (e.g., west or east coast city),
- language/script (e.g., English vs. Chinese),
- more vowels vs. more consonants, or sequence identity in a bimodal corpus (e.g., choose AAAA vs. BBBB),
- syntactic pattern (e.g., subject-verb-object (SVO) vs. subject-object-verb (SOV); singular vs. plural agreement).

This breaks for two reasons: (1) We cannot hand-enumerate all relevant linguistic factors; it is expensive, incomplete and defeats the purpose of learning structure from data. (2) At inference the latents are unknown; naive independent sampling of $z$ ignores multi-modality and correlations between factors, producing inconsistent outputs.

These issues motivate *learning* both the sequence-level latent(s) and their distribution end-to-end, which we address with autoencoders (AEs), specifically variational autoencoders (VAEs).

## 2.3 AUTOENCODERS

A deterministic autoencoder introduces a learned encoder $f_\phi$ and decoder $g_\theta$:

$$x_{1:T} \xrightarrow{f_\phi} z \xrightarrow{g_\theta} \hat{x}_{1:T},$$

and minimizes reconstruction loss (token-level cross-entropy for text):

$$\mathcal{L}_{\text{AE}}(x) = -\sum_{t=1}^{T} \log p_\theta(x_t \mid f_\phi(x), t). \tag{4}$$

---

[1]The attention KV cache typically scales as $\mathcal{O}(L\,T\,d)$ per sequence (model width $d$); large $T$ or large batches trade memory for throughput.

This addresses the first issue, we learn the required features (language, topic, style, word order..) to help the decoder reconstruct, without hand-specifying factors.

However, two problems remain:

**(i) No calibrated prior.** At inference we need a rule to sample $z$, but plain AEs learn latents with an unknown, highly non-Gaussian data distribution; naive sampling from an ad-hoc prior can produce off-manifold latents and incoherent outputs.

**(ii) Uncontrolled capacity.** If $z$ is too small, reconstruction fails. If $z$ is high-dimensional, the encoder can pass through the entire sequence unchanged, and both encoder and decoder collapse to identity functions. Generic regularizers (dropout/weight decay) do not provide an explicit information budget.

These motivate a probabilistic latent model with (a) a learnable *posterior* for $z$ and (b) an explicit *prior* that enables sampling and controls capacity.

## 2.4 Variational Autoencoders

A VAE introduces a distribution over latent variables $z$ to model a sequence $x_{1:T}$ and maximizes the evidence lower bound (ELBO):

$$\text{ELBO}(x) = \mathbb{E}_{q_\phi(z|x)}[\log p_\theta(x \mid z)] - D_{\text{KL}}\big(q_\phi(z \mid x) \,\|\, p(z)\big), \tag{5}$$

with prior $p(z) = \mathcal{N}(0, I)$. The encoder defines a Gaussian posterior

$$q_\phi(z \mid x) = \mathcal{N}\big(\mu_\phi(x), \, \text{diag}(\sigma_\phi^2(x))\big), \quad z = \mu_\phi(x) + \sigma_\phi(x) \odot \epsilon, \;\; \epsilon \sim \mathcal{N}(0, I), \tag{6}$$

and the decoder $p_\theta(x \mid z)$ scores $x$ conditioned on $z$ (autoregressively in classic text VAEs; non-autoregressively in VALM). Training minimizes the negative ELBO, i.e., token-level reconstruction (cross-entropy in practice) plus a KL regularizer.

**Inference.** Unconditional generation samples $z \sim p(z)$ and decodes once from $p_\theta(x \mid z)$. Conditional variants use $z \sim q_\phi(z \mid x_{\text{obs}})$.

**Intuition.** The KL term controls the information capacity of the latent bottleneck: it discourages $q_\phi(z \mid x)$ from carrying a code that directly passes through the uncompressed $x$, and the reparameterization $z = \mu + \sigma \odot \epsilon$ injects noise that encourages smooth, generative latents. In classic text VAEs with strong AR decoders this can lead to *posterior collapse* (the decoder ignores $z$); VALM removes the AR path so the decoder must use $z$ to model sequence-level dependencies.

## 2.5 Application to text: VALM

VAEs are most common in continuous domains such as images and audio; here we apply the same probabilistic latent approach to *discrete* text and decode in a single pass [2]. We train with Alg. 1 and generate with Alg. 2.

**Architectural choice.** In all experiments the *encoder* and *decoder* are bidirectional Transformers, with no causal masking and no autoregressive components. This contrasts with prior text-VAEs that keep an AR decoder and use $z$ only as a global control signal, retaining sequential inference and enabling the decoder to ignore $z$ (posterior collapse). VALM removes the causal path entirely: decoding depends on $z$ only, so the latent must carry sequence-level information and inference is fully parallel.

**Decoder and $z$ injection.** The decoder is a bidirectional Transformer that conditions on $z$ and absolute positions to model

$$p_\theta(x \mid z) = \prod_{t=1}^{T} p_\theta\big(x_t \mid z, t\big), \tag{7}$$

emitting per-position logits in a *single* forward pass.

---

[2] We maintain continuous latents; inputs and outputs are discrete tokens.

To condition on $z$, we project it to the model width and broadcast to all positions:

$$e_t^{(0)} = \underbrace{E[x_t]}_{\text{token embed}} + \underbrace{P[t]}_{\text{positional}} + \underbrace{W_z z + b_z}_{\text{global latent bias}}, \quad W_z z + b_z \in \mathbb{R}^{d_{\text{model}}}. \tag{8}$$

**Complexity and sequential cost** Let $T$ be sequence length and $L$ the number of Transformer layers. AR decoding performs $T$ dependent steps: $\mathcal{O}(LT)$. VALM computes all logits in one pass: $\mathcal{O}(L)$. This comparison concerns sequential depth (latency), not total arithmetic: for a fixed architecture and sequence length, per-sample FLOPs per pass are on the same order; VALM reduces the number of dependent steps, not the overall amount of compute. These depth statements hold for a fixed architecture; matching a target quality may require different $L$ or width across AR and VALM.

## 2.6 $\beta$-ADJUSTED VAEs

We optionally weight the KL term:

$$\mathcal{L}(x) = -\mathbb{E}_{q_\phi(z|x)}[\log p_\theta(x \mid z)] + \beta \cdot D_{\text{KL}}\big(q_\phi(z \mid x) \,\|\, p(z)\big), \tag{9}$$

but keep to $\beta = 1$ by default.

We anneal $\beta$ linearly from 0 to 1 over the first 15% of steps:

$$\beta_t = \min\left(1, \frac{t}{0.15\,T}\right). \tag{10}$$

## 2.7 PADDING AND VARIABLE LENGTH

We train on fixed-length blocks of $T_{\max}$ tokens. Each sequence $x$ is truncated or right-padded with `<pad>` to length $T_{\max}$. The decoder always outputs $T_{\max}$ logits; at inference we detokenize and drop `<pad>` tokens.

**Loss masking.**

$$\mathcal{L}_{\text{rec}}(x, z) = -\frac{1}{\sum_t m_t} \sum_{t=1}^{T_{\max}} m_t \, \log p_\theta(x_t \mid z, t), \quad m_t = \mathbf{1}[x_t \neq \texttt{<pad>}]. \tag{11}$$

## 2.8 TEMPERATURE

AR LMs typically control diversity with *logit temperature* (token-level softmax scaling). In VALM, per-token sampling breaks global consistency because all positions are decoded independently given a single $z$. We therefore control diversity by scaling the *latent* instead.

**Definition.** With prior $p(z) = \mathcal{N}(0, I)$, introduce a latent temperature $\tau > 0$ at inference and sample

$$z \sim \mathcal{N}(0, \tau^2 I) \quad \Longleftrightarrow \quad z = \tau\,\epsilon, \ \epsilon \sim \mathcal{N}(0, I). \tag{12}$$

This preserves global coupling: one $z$ drives all positions in a single pass. $\tau = 0$ yields a deterministic output (fixed $z$); increasing $\tau$ increases diversity at the cost of inference coherence.

## 2.9 SUMMARY

VALM replaces the autoregressive path with a single global latent $z$ and a bidirectional decoder that emits all token logits in one pass. Training maximizes the ELBO with masked reconstruction (pads ignored) and a KL regularizer; inference samples $z$ from the prior and decodes once. Order is preserved by absolute positional conditioning, and sampling uses a standard Gaussian prior $p(z) = \mathcal{N}(0, I)$. Latent temperature (Sec. 2.8) modulates global diversity at inference. Unless stated otherwise, VALM is *unconditional*.

---

**Algorithm 1** VALM training (single pass)

---

1: **Input:** batch $x$ (padded/truncated to $T_{\max}$), positions $t=1{:}T_{\max}$, mask $m_t=\mathbf{1}[x_t \neq \texttt{<pad>}]$
2: $(\mu, \sigma) \leftarrow \mathrm{Enc}_\phi(x)$
3: Sample $\epsilon \sim \mathcal{N}(0, I);\quad z \leftarrow \mu + \sigma \odot \epsilon$
4: logits $\leftarrow \mathrm{Dec}_\theta(z, \text{positions})$              $\triangleright$ parallel per-token logits
5: **Reconstruction:** $\quad \mathcal{L}_{\text{rec}} = -\dfrac{1}{\sum_t m_t} \sum_{t=1}^{T_{\max}} m_t \, \log p_\theta(x_t \mid z, t)$
6: **KL:** $\quad \mathcal{L}_{\text{KL}} = D_{\text{KL}}\big(q_\phi(z \mid x) \,\|\, p(z)\big)$
7: **Total loss:** $\quad \mathcal{L} \leftarrow \mathcal{L}_{\text{rec}} \; + \; \beta \, \mathcal{L}_{\text{KL}}$
8: Update $\phi, \theta$ with AdamW

---

**Algorithm 2** VALM inference (single pass)

---

1: **Unconditional:** sample $z \sim \mathcal{N}(0, \tau^2 I)$    with latent temperature $\tau$ (see Sec. 2.8; default $\tau = 1$)
2: logits $\leftarrow \mathrm{Dec}_\theta(z, \text{positions})$
3: $\hat{x}_t \leftarrow \arg\max \text{logits}_t$
4: Detokenize; drop $\texttt{<pad>}$ tokens

---

## 3 EXPERIMENTS

### 3.1 VALM-1 MODEL

To test if VALM can produce coherent text in one pass, we train VALM-1.

**Hyperparameters** VALM-1 has `d_model=1024`, `n_layers=16` and 420M overall parameters divided between 201M encoder, 201M decoder, 16.8M embedding and 0.1M latent-projection parameters. We train with $\beta = 2.0$ for 1500 epochs. Where not stated otherwise, we use the hyperparameters listed in Appendix B.

**Dataset** We use only the *passage* field of the bAbI dataset for unconditional modeling; the *question* and *answer* fields are discarded. VALM-1 is trained purely for unconditional generation; no QA tasks are used.

| id | bAbI passage excerpts | | |
|---|---|---|---|
| A | Mary went back to the bedroom.
Daniel travelled to the office.
Daniel went back to the garden.
John journeyed to the office. | B | Daniel is in the office.
Daniel journeyed to the garden.
John is in the office.
Daniel went back to the bedroom. |

Table 1: Manually selected bAbI "passage" lines used for qualitative style reference. Truncated to 32 tokens to match the model's tokenizer.

**Single-pass coherence.** At $\tau = 0.5$, the random samples in Table 2 read as coherent bAbI-style passages despite being produced in one forward pass that emits all 32 tokens in parallel. The model follows the dataset's line-by-line event template (short SVO or copular sentences) in e.g. Table 2. It keeps short-horizon entity and object links: id=1 threads "Antoine is tired" → "Antoine moved to the bedroom" → "Antoine took the pajamas there"; id=4 keeps the thirst-milk link ("Sumit is thirsty" → "Sumit took the milk there"); id=5 maintains the apple across transfers ("Mary took the apple there" → "Mary passed the apple to John" → "John handed the apple to Mary"). These patterns indicate local role consistency and event chaining from a single global latent, not from autoregressive conditioning.

---

[3]No manual selection or sorting; same decoding parameters across rows.

| id | text | id | text |
|---|---|---|---|
| 1 | Antoine is tired.
Antoine moved to the bedroom.
Antoine took the pajamas there.
Jason is bored ... | 2 | Sandra journeyed to bedroom.
Mary journeyed to the bedroom.
Sandra got the football there.
Sandra put down ... |
| 3 | Sandra journeyed to the office.
Mary journeyed to the bathroom.
Sandra got the football.
John moved to the garden. ... | 4 | Sumit is thirsty.
Sumit took the milk there.
Antoine is tired.
Sff journeyed to the office. ... |
| 5 | Daniel travelled to the office.
Mary took the apple there.
Mary passed the apple to John.
John handed the apple to Mary.
John ... | 6 | Mary and Sandra moved to the kitchen.
Sandra and Mary journeyed to the hallway.
Daniel and Sandra went back to ... |
| 7 | Sumit is thirsty.
Antoine is thirsty.
Jason is thirsty.
Yann is tired.
Yann journey ... | 8 | Sandra journeyed to the bathroom.
John journeyed to the kitchen.
Sandra journeyed to the hallway.
Daniel journeyed ... |
| 9 | Bill moved to the bedroom.
Jeff went to the hallway.
Jeff travelled to the bedroom.
Bill went to the office. ... | 10 | Mary travelled to the bedroom.
Mary is in the garden.
John is in the bathroom.
Daniel is in the bedroom.
Mary moved to ... |

Table 2: Random VALM-1 single-pass generations ($\tau = 0.5$).[3]

**Limits and errors.** Yet the model sometimes generates the wrong token at a single position (e.g., "Sandra journeyed to bedroom" instead of "Sandra journeyed to the bedroom") or a corrupted name ("Sff" instead of "Jeff"). It also has verb-form errors ("Yann journey"), and occasional state mismatches within a sample (id=10: "Mary travelled to the bedroom" followed by "Mary is in the garden"). Several endings are truncated mid-event, which is due to the fixed 32-token output limit rather than a loss of coherence.

The per-token KL divergence reached 0.4 at the end of training, which corresponds to approximately 0.58 bits per token. This indicates the model uses the latent to pass information while still compressing it; under KL collapse it would be near zero. Training curves are shown in Appendix D.

### 3.2 SCALING LAWS

To study the performance of VALM, we trained **124** VALM variants on *TinyStories* (Eldan & Li, 2023) and **45** VALM runs on *WikiText-103* (Merity et al., 2016). We fit a two-factor power law

$$\mathcal{L}(N, P) \approx c + a_t N^{-\alpha_t} + a_p P^{-\alpha_p}, \tag{13}$$

to the validation (TinyStories) and train (WikiText-103)[4] losses, respectively.

| Dataset | runs | $\alpha_t$ | $\alpha_p$ | $c$ | $a_t$ | $a_p$ | $R^2$ |
|---|---|---|---|---|---|---|---|
| TinyStories (VALM) | 124 | 0.27 | 0.32 | 2.11 | 68.31 | 83.37 | 0.83 |
| WikiText-103 (VALM) | 45 | 0.09 | 0.32 | 0.88 | 13.78 | 49.93 | 0.95 |
| Chinchilla AR (ref.) | 400 | 0.34 | 0.28 | 1.69 | 406.40 | 410.70 | - |

Table 3: Combined scaling fits: VALM on TinyStories and WikiText-103 (Merity et al., 2016), and AR reference exponents from Chinchilla (Hoffmann et al., 2022).

These results are for VALM's ELBO objective; absolute constants are not directly comparable to AR cross-entropy fits, but the exponents are informative, showing that token and parameter sensitivity

---

[4]Unfortunately, due to a software bug the validation losses weren't recorded for these runs.

follow similar orders to strong AR baselines and support near-balanced compute splits, albeit with a weaker scaling result on tokens for WikiText-103 VALM. We generally observed no KL collapse; issues appeared only under extreme hyperparameters (see Appendix E.2).

### 3.3 Ablation with naive no-encoder model

We run an ablation with no encoder and a fixed, learnable latent. Loss is much higher (train $0.66 \rightarrow 3.55, +2.90$; val $1.62 \rightarrow 3.43, +1.81$) and samples degrade to frequent tokens and punctuation. See Appendix E.3 for details and examples.

## 4 Related Work

### 4.1 Autoencoder with Autoregressive components

**AR decoders conditioned on a latent (posterior collapse)** Most text VAEs keep an autoregressive decoder and use $z$ as a weak control signal (Bowman et al., 2016; Zhao et al., 2017; Wen et al., 2017; Li et al., 2020; Kaiser et al., 2018a). This often collapses the posterior under teacher forcing because the AR path explains the data without using $z$ (He et al., 2019; Fu et al., 2019). Variants with RNN/CNN decoders or auxiliary/local AR losses retain the same failure mode (Yang et al., 2017; Zhang et al., 2017). *VALM:* removes the AR path and decodes all positions in one pass from a single global $z$.

**Human-readable latent texts / discrete bottlenecks** Compressing to discrete or human-readable latents adds discrete training (REINFORCE, Gumbel, codebooks) and usually decodes autoregressively from the latent text (Kaiser et al., 2018a; Li et al., 2020). *VALM:* continuous Gaussian latent, no AR component.

**Discrete latents and vector quantization** VQ models compress to short code sequences and often train a separate AR prior over codes (Kaiser et al., 2018b; Razavi et al., 2019). *VALM:* no AR prior at any level.

### 4.2 Other

**Diffusion / masked language model (MLM) / iterative refinement** Masked-LM and refinement models generate by repeatedly masking and filling or by denoising over multiple steps (Lee et al., 2018; Ghazvininejad et al., 2019; Wang et al., 2021). They require a schedule (mask ratio or noise level) and $K > 1$ passes at inference; even "constant time" CMLM uses a fixed small number of refinement steps rather than a single pass (Ghazvininejad et al., 2019). Commercial diffusion systems report high throughput but still use step-by-step refinement and do not release weights (Labs et al., 2025; DeepMind, 2024) VALM does not iterate and we publish architectural details.

**One-step decoders vs. VALM** The inference-time decoder of a one-step *distilled diffusion* model, a *GAN*, or *Parallel WaveNet* looks like VALM (single latent$\rightarrow$ per-position logits in one pass). However, training them requires: (1) a teacher and denoising/score-matching objective with noise schedules and often guidance (distilled diffusion) (Lee et al., 2018; Ghazvininejad et al., 2019; Yin et al., 2024; Xie et al., 2024; Song et al., 2023; Chen et al., 2025); (2) an adversarial discriminator with RL/continuous relaxations for discrete tokens, yielding unstable dynamics that trail MLE baselines (GANs) (Yu et al., 2017; Caccia et al., 2020; Ren et al., 2023); (3) an autoregressive teacher and invertible flow constraints with probability density distillation (Parallel WaveNet) (van den Oord et al., 2017). *VALM* instead uses plain likelihood (ELBO) with a calibrated Gaussian prior: no teacher, no guidance, no discriminator, no invertibility/flow constraints, and no AR path.

**Non-autoregressive translation (NAT/NAR)** Similar to a naive parallel approach, but translation restricts outputs to a target language. There are still many valid translations for a single source sentence, hence the "multimodality problem"[5]. NAT systems add extra signals or procedures-word

---

[5]Here "multimodality" means one-to-many target ambiguity in translation: multiple valid target sequences for the same source. It does not refer to multimodal inputs like text+image.

fertilities, CTC, auxiliary regularizers, or iterative refinement-and often predict length explicitly (Gu et al., 2018; Libovický & Helcl, 2018; Shu & Nakayama, 2019; Saharia et al., 2020; Gu & Kong, 2021; Li et al., 2022; Zhou et al., 2022). In contrast, VALM is unconditional with (i) no iterative refinement, (ii) no separate length module - padding handles variable length, and (iii) a single global latent that binds sequence-level choices.

**Specialized hardware**   Accelerators such as Cerebras and Groq reduce AR latency by hardware speedups but keep the $O(TL)$ sequential dependency (Cerebras Systems, 2021; Groq, Inc., 2024). *VALM:* removes the token-by-token dependency algorithmically.

### 4.3 AUTOENCODERS AND FLOWS

**Bag-of-Words (BoW) decoders and plain autoencoders**   BoW decoders drop order and are not viable sequence generators (Miao et al., 2016; Srivastava & Sutton, 2017). Deterministic autoencoders reconstruct but lack a calibrated prior, so sampling is off-manifold (Zhang et al., 2017; 2018; Montero et al., 2021). *VALM:* preserves order via positional conditioning and samples from a simple Gaussian prior.

**Plain (non-variational) convolutional autoencoders**   Deterministic CNN autoencoders decode non-autoregressively for reconstruction; related set-prediction decoders target selection rather than unconditional generation (Zhang et al., 2017; 2018; UCSD NLP Group, 2017; Cheng et al., 2023). *VALM:* probabilistic latent with an ELBO objective and single-pass generation.

**Normalizing flows**   FlowSeq and IAF-style models can generate in one shot but require invertibility, Jacobian terms, or categorical/argmax relaxations, adding architectural and training cost (Kingma et al., 2016; Ziegler & Rush, 2019; Ma et al., 2019; Su et al., 2020; Hoogeboom et al., 2021). *VALM:* standard non-invertible transformers with no log-det terms.

## 5 LIMITATIONS AND FUTURE WORK

**Scope and scale.** Our experiments are small and on modest corpora; results may not transfer to larger vocabularies, longer contexts, or diverse domains. Future work can evaluate larger models on larger datasets and longer sequences.

**Unconditional decoding and limited control.** VALM-1 decodes from a single global latent $z$ without conditioning on a prompt, so it cannot continue a prefix, answer questions, or enforce constraints. There is no mechanism for length, topic, or style control, and no safety filtering during decoding. Token-level temperature is ineffective under our factorization; only latent temperature controls diversity, and only coarsely. Future work can introduce conditioning for more practical use cases.

## 6 CONCLUSION

We introduced VALM, a non-autoregressive variational language model that emits all token logits in a single pass from a global latent. In small-scale experiments, VALM-1 produces coherent 32-token spans and shows predictable improvements with model/data scale under an ELBO objective. Removing the autoregressive path forces the decoder to use the latent, which mitigated posterior-collapse issues we observed only under extreme hyperparameters, and showing the applicability of pure VAEs to text. While current results are unconditional and short-context, they indicate that single-pass generation is viable; future work will add conditioning, extend context, and evaluate larger models on broader corpora.

**Ethics Statement   Data provenance and licensing.** We train exclusively on public datasets (bAbI *passages* only, TinyStories, WikiText-103). We did not scrape private sources or collect new data.

**Dual-use and misuse.** Standard risks of spam, bias, and toxicity persist. Faster, high-throughput generation can also make it harder for users to verify outputs in real time, increasing the chance of unvetted content spreading. VALM-1 is a small, unconditional research model and is *not* intended

for production use. For any future, larger or fine-tuned variants, we recommend rate limiting, content filtering, abuse reporting channels, and provenance measures (e.g., optional watermarking) to reduce misuse.

**Reproducibility Statement**   We release code, configuration files, and experimental scripts to reproduce training and evaluation as an anonymous repository. The configuration files contain all hyperparameters and random seeds.

**Acknowledgments**   Left empty during peer review.

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

# A APPENDIX

# B HYPERPARAMETERS

We use symmetric encoder and decoder transformer stacks with the same size and hyperparameters. We use no label smoothing.

## B.1 DATASETS & PREPROCESSING

**Corpora.** We use bAbI (passages only), TinyStories, and WikiText-103 via Hugging Face Datasets: `Muennighoff/babi`, `skeskinen/TinyStories-hf`, and `wikitext-103-raw-v1`. For bAbI we rename `passage` to `text` and discard `question`/`answer` (unconditional modeling only).

**Splits.** We rely on the datasets' provided `train`/`validation` splits; we do not use a test split. When sub-sampling is requested (HPO sweeps), we select the first `train_samples` and `test_samples` examples from the respective splits after shuffling. Training batches are shuffled; random seeds (e.g., 42) are set in configs for reproducibility.

Table 4: Default hyperparameters

| Category | Value |
|---|---|
| *Data / Tokenization* | |
| Sequence length $T_{\max}$ | 32 |
| Tokenizer | Custom BPE tokenizer trained on TinyStories |
| Vocab size | 8196 |
| *Model (Transformer)* | |
| $d_{\text{model}}$ | 1024 |
| Layers (Enc/Dec) | 16 / 16 |
| Heads (Enc/Dec) | 8 / 8 |
| Feedforward $d_{\text{ff}}$ | $4 \times d_{\text{model}} = 4096$ |
| Activation (FFN) | GELU |
| Positional encodings | Absolute (learned); RoPE/ALiBi not used |
| Normalization | Pre-norm (LayerNorm) |
| Dropout | 0.0 (disabled) |
| Implementation | built-in PyTorch Transformer |
| *Latent $z$* | |
| Global latent dim $d_z$ | 32 |
| $\beta$ (ELBO weight) | 1.0 |
| $\beta$ warmup ratio | 0.15 (enabled) |
| Latent temperature $\tau$ (inference) | 1.0 (default) |
| *KL term uses standard weighting $\beta$; no free-bits.* | |
| *Optimization / Training* | |
| Optimizer | AdamW |
| Adam $(\beta_1, \beta_2)$ | (0.95, 0.95) |
| Peak LR | $8 \times 10^{-4}$ |
| LR schedule | Cosine decay to zero with linear warmup |
| LR warmup ratio | 0.10 |
| Batch size | 512 |
| Grad. accumulation | 1 |
| Weight decay | 0.0 (disabled) |
| Gradient clipping | Disabled |
| Precision | `16-mixed` |
| Activation checkpointing | Enabled at residual vectors of each layer |
| *Other* | |
| seed | 42 |

**Filtering.** Before tokenization we drop empty or whitespace-only rows; we apply no additional deduplication or cleaning.

**Tokenizer.** A custom BPE tokenizer trained on TinyStories (vocab size 8196) with a pad token `<pad>`. We tokenize with truncation and right-padding to a fixed maximum length `seq_len= 32`, and we keep only `input_ids` (no attention masks).

**Note on conditioning.** Some VAEs for text and audio include side information $c$ and model $p(x \mid z, c)$ (e.g., prompts, class labels, or transcripts). In this work we restrict to *unconditional* VAEs: no prompts or auxiliary conditioning-$z$ is the only context variable. As a result, generations are random sentences from the overall training distribution, and utility is limited by short context and small datasets; our aim here is to demonstrate feasibility and provide a transparent baseline. Future work will add conditioning for controlled generation.

| Name | Type | Params | Mode |
|------|------|--------|------|
| model | VALM | 420 M | train |
| model.tok_emb | Embedding | 8.4 M | train |
| model.encoder | TransformerEncoder | 201 M | train |
| model.to_mu | Linear | 32.8 K | train |
| model.to_logvar | Linear | 32.8 K | train |
| model.z2dec | Linear | 32.8 K | train |
| model.decoder | TransformerEncoder | 201 M | train |
| model.output_fc | Linear | 8.4 M | train |

Table 5: VALM-1 detailed parameter count.

## C  VALM-1 DETAILED PARAMETER COUNT

### C.1  LATENT HEADS AND Z2DEC SETUP

**Latent heads.** The encoder produces a single pooled vector $h_{\text{enc}} \in \mathbb{R}^{d_{\text{model}}}$ (see Fig. 1). Two linear heads map this to the mean and log-variance of a diagonal Gaussian posterior:

$$\mu = W_\mu \, h_{\text{enc}} + b_\mu, \qquad\qquad W_\mu \in \mathbb{R}^{d_z \times d_{\text{model}}},$$
$$\log \sigma^2 = W_{\text{logvar}} \, h_{\text{enc}} + b_{\text{logvar}}, \qquad\qquad W_{\text{logvar}} \in \mathbb{R}^{d_z \times d_{\text{model}}}.$$

We use $q_\phi(z \mid x) = \mathcal{N}\big(\mu, \, \text{diag}(\sigma^2)\big)$ with reparameterization $z = \mu + \sigma \odot \epsilon, \, \epsilon \sim \mathcal{N}(0, I)$ and $\sigma = \exp\big(\frac{1}{2} \log \sigma^2\big)$.

**z2dec projection.** A linear map projects $z \in \mathbb{R}^{d_z}$ to the model width and adds a global bias to every position in the decoder input (Eq. 7):

$$h_z = W_z z + b_z \in \mathbb{R}^{d_{\text{model}}}, \qquad e_t^{(0)} = E[x_t] + P[t] + h_z.$$

Here $W_z \in \mathbb{R}^{d_{\text{model}} \times d_z}$ and $b_z \in \mathbb{R}^{d_{\text{model}}}$. This module corresponds to `model.z2dec` in Table 5.

**Initialization.** Unless noted otherwise, linear layers use the PyTorch defaults for weights, and all projection biases are initialized to $0.0$ (`to_mu`, `to_logvar`, and `z2dec`).

## D  TRAINING CURVES FOR VALM-1

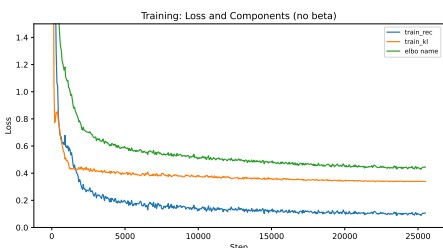

Figure 2: ELBO loss and components.

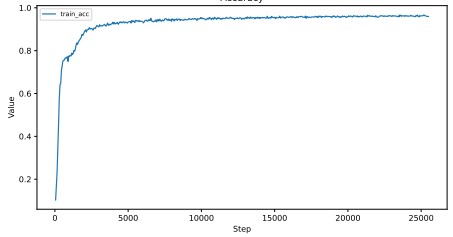

Figure 3: Word accuracy: During training what percent of tokens had the correct token as the most likely predicted by the model.

| id | text | id | text |
|---|---|---|---|
| 0 | Mice are afraid of cats.
Wolves are afraid of mice.
Sheep are afraid of mice.
Emily is a sheep.
W ... | 1 | Theiel is is of the red.
Theand is is of the the.
The went is theang the.
The moved is the ... |
| 2 | The hallway is east of the bedroom.
The bedroom is east of the.
The to is north of the to.
The is ... | 3 | Sandra picked up the milk there.
John went to the hallway.
Mary moved to the office.
Sandra dropped the milk. ... |
| 4 | The hallway is north suitcase the garden.
The bigger is n boxh chocolates.
The chestThe bigger than the container.
The box ... | 5 | Sandra travelled to the kitchen.
John is in the bedroom.
Mary is not in the hallway.
Mary is in the bathroom.
Mary ... |
| 6 | Billats are took the football there.
J ff gave the football to Fred.
Fred moved to the kitchen.
Bill gave are afraid to the ... | 7 | The bedroom is south of the garden.
The bathroom is south of the hallway.
The office is west of the hallway.
The bedroom is ... |
| 8 | Mary and John went back to the kitchen.
After that they went back to the garden.
Daniel and Daniel went to the bathroom.
Mary ... | 9 | Mary travelled to the office.
Sandra moved to the kitchen.
Mary grabbed the milk there.
John got the apple there.
Sand ... |

Table 6: Random VALM-1 single-pass generations at latent temperature $\tau = 1.0$. More errors than at $\tau = 0.5$ (cf. Table 2).[6]

| | count | mean | std | min | 5% | 25% | 50% | 75% | 95% | m |
|---|---|---|---|---|---|---|---|---|---|---|---|
| params | 124 | 5.77e+07 | 3.52e+07 | 98,304 | 393,216 | 3.07e+07 | 5.66e+07 | 8.49e+07 | 1.01e+08 | 2.01e+ |
| tokens | 124 | 5.65e+06 | 3.26e+07 | 8,096 | 100,132 | 100,132 | 131,072 | 524,288 | 2.12e+06 | 2.12e+ |
| flops | 124 | 1.28e+14 | 1.57e+14 | 1.25e+12 | 5.49e+12 | 2.96e+13 | 6.05e+13 | 1.78e+14 | 5.34e+14 | 6.23e+ |

Table 7: Summary statistics for the TinyStories scaling law.

# E    SCALING LAW FIT

## E.1    SUMMARY STATS (PARAMS, TOKENS, FLOPs)

## E.2    KL COLLAPSE FOR EXTREME HYPERPARAMETERS

In rare settings with extreme hyperparameters we observed a failure mode where the encoder posterior stays close to the prior across examples (i.e., $\mu \approx 0$, $\sigma \approx 1$), effectively minimizing the KL term while leaving the reconstruction loss large. This is a suboptimal local minimum in which the model does not use the latent to encode information.

This differs from classic posterior collapse in text VAEs with powerful autoregressive decoders, where the decoder can explain the data without $z$ and collapse can be optimal for that objective. In VALM, decoding depends on $z$, so collapse arises only from optimization pathologies or extreme settings.

When did it occur? Only under extreme choices such as very small weight-initialization scale, very deep stacks (e.g., 32 layers), or very large $\beta$ in $\beta$-VAE. Mitigations that helped in our tests include mild $\beta$ warmup, standard initialization scales, and moderate depth.

---

[6]No manual selection or sorting; same decoding parameters across rows.

### E.3    ABLATION: NO ENCODER - FIXED LEARNABLE LATENT

We remove the encoder and replace $q_\phi(z \mid x)$ with a single trainable vector $\mathbf{z}_\star$ that is shared across all examples. Training minimizes reconstruction only; the KL term is zero by construction because $z$ is deterministic and global. At inference we decode once from $\mathbf{z}_\star$.

**Outcome.**    Validation loss increases, word accuracy decreases, and samples collapse toward a single passage template with small token-level variations. This confirms that learning a posterior that depends on $x$ is necessary even for short spans.

| Task | Setting | Val loss | Val acc | KL/token |
|------|---------|----------|---------|----------|
| bAbI (passages) | VALM-1 baseline | 1.6229 | - | ≈0.58 bits |
| bAbI (passages) | No encoder, $\mathbf{z}_\star$ | 3.4331 | - | 0.00 |
| TinyStories | VALM-1 baseline | - | - | ≈0.58 bits |
| TinyStories | No encoder, $\mathbf{z}_\star$ | - | - | 0.00 |

Table 8: Ablation removing the encoder and using a single trainable global latent $\mathbf{z}_\star$.

**bAbI numbers (seq=32).**    Baseline: train= 0.6550, val= 1.6229. No-encoder: train= 3.5543, val= 3.4331. Deltas: +2.8993 train, +1.8102 val. Samples from the no-encoder variant were highly repetitive (e.g., "The and is the the the .."), indicating collapse toward a single template.

| id | baseline (temp=1.0) | id | no encoder (temp=1.0) |
|----|---------------------|----|-----------------------|
| 1 | Bill went back the the this there. \\Yesterday Fred journey to to bedroom office. \\Bill morning travelled the the school. \\Yesterdayed to the | 1 | Theand is the the the.. \\the the... \\the. the.. \\the the.. |
| 2 | John and isandra \\to the garden. \\After that theyumit the kitchen. \\Johnason journey moved to the garden. \\After that they | 2 | Theand is the the the.. \\the the... \\the. the.. \\the the.. |

Table 9: Qualitative comparison at $\tau = 1.0$ (seq=32). The no-encoder variant collapses to high-frequency tokens and punctuation, while the baseline retains event-like structure despite errors. Notation: \\denotes a line break.

**Why this happens.**    With no encoder, the decoder receives no information about the input $x$. Under cross-entropy, the best it can do is predict corpus-level frequent tokens at each position. On bAbI that means tokens like "the" and periods; hence the repetitive fragments above.

### E.4    WHY PAD (NOT EOS).

VALM decodes in one pass, so no explicit `<eos>` is needed. Since positions are conditionally independent given $z$, an `<eos>` at one index would not suppress later logits; masked padding is simpler and avoids length bias.

## F    OTHER NOTES

### F.1    USE OF LLMs

LLMs were used to (1) help write experimental code and scripts and (2) correcting spelling and formatting in the final version of the paper.

### F.2    SCOPE AND NEXT STEPS.

We did not use prompts/conditioning, did not train for $> 32$ tokens, and did not compare against strong autoregressive baselines-our goal here was to introduce the method, demonstrate feasibility

of single-pass text generation and provide a transparent recipe. Next, we will add conditioning, extend context length, and evaluate larger models on broader corpora and tasks.

