# OpenReview forum: "VALM: Variational Autoencoder Language Models for Highly Parallel Text Generation"
_ICLR.cc/2026/Conference — ICLR 2026 Conference Withdrawn Submission_

### Official Review · Reviewer_CNzG · 2025-10-28

**Soundness:** 1
**Presentation:** 2
**Contribution:** 1
**Rating:** 2
**Confidence:** 3

**Summary:**

This is basically asking - can we making a VAE-like model which instead of the classical token-by-token transformer setup, predicts every token in parallel at once ? Such attempts have been made in general, the paper notes diffusion models but the idea of using a latent to make > 1 prediction steps is not new. The usage of a VAE is somewhat unusual in the context.

To be honest, I am very skeptical of the idea, but in any case, the idea is not well executed (in my opinion). I detail on this below.

**Strengths:**

The idea of a fully parallel (for all tokens) generation process with a VAE is somewhat novel, and I like that there are some attempts to look at the scaling law part of it (although it seems to be very randomly inserted)

**Weaknesses:**

Unfortunately, the paper, in my opinion, suffers from multiple flaws.

1 - On a theoretical level, fully parallel generation may not even be efficient. One can think of infinite sentences that convey the same meaning.  Therefore, latent space that only maps a latent to an idea, and then autoregressively and non-deterministically generates the words to convey that meaning, requires less work that mapping separate latents to all sentences that mean exactly the same thing. Fully parallel generation may stumble in these scenarios, e.g. consider conveying the meaning of "A is shorter than B" or "B is taller than A". Assuming that these are the only two representations for simplicity, first token + latent that maps to meaning can auto-regressively generate it, but the two statements must have different latents in the fully parallel world.

2 - Beyond the theoretical problems, there is virtually no empirical work. There is no good empirical study on a lot of benchmarks against meaningful models, use of pre-trained encoders or decoders to see if things make sense, no study of latent space characteristics beyond some simple generated sentences...

**Questions:**

I simply ask for any one summary comparison table against peer methods (you have cited quite a few - and yet - where is the comparison table ?)

---

### Official Review · Reviewer_xHKq · 2025-10-31

**Soundness:** 1
**Presentation:** 1
**Contribution:** 1
**Rating:** 0
**Confidence:** 3

**Summary:**

Autoregressive language models (LMs) have demonstrated impressive capabilities across various domains. However, their token-by-token decoding inherently limits inference speed. This paper investigates the integration of Variational Autoencoders (VAEs) into a non-autoregressive architecture that predicts entire sentences in parallel from a single global latent variabe. Experimental results provide preliminary evidence of the effectiveness of the proposed approach.

**Strengths:**

- Research on introducing VAEs into encoder-decoder architectures for fast inference (i.e., a single forward pass) has the potential to benefit the community.

**Weaknesses:**

### Methodology:

- Introducing VAEs into encoder-decoder architectures for fast inference is not particularly novel. As mentioned in the related work, VAE-based non-autoregressive models are highly relevant and widely used in machine translation tasks (conditional generation). Unfortunately, the paper does not clearly highlight the differences between the proposed approach and the existing VAE-based non-autoregressive models. From my understanding, the main difference appears to be limited to architectural modifications.

---

### Experiments:

- The paper does not include any baselines, such as standard VAE-based autoregressive LMs, VAE-based non-autoregressive models, or diffusion LMs. Without a comparison to the most relevant VAE-based non-autoregressive models, the evaluation is not convincing. It seems reasonable that they could be adapted for unconditional generation tasks as well. At a minimum, the authors should attempt to include such a baseline [1] and present corresponding results.

- No evaluation metrics commonly used in language modeling and generation [2] are reported. The paper only presents generated sentences, making it difficult to quantitatively assess the performance or effectiveness of the proposed approach.

> **Reproducibility Statement:** We release code, configuration files, and experimental scripts to reproduce training and evaluation as an anonymous repository. The configuration files contain all hyperparameters and random seeds.

- However, I was unable to locate the anonymous repository referenced in the Reproducibility Statement.

[1] Latent-Variable Non-Autoregressive Neural Machine Translation with Deterministic Inference Using a Delta Posterior.

[2] LlaMaVAE: Guiding Large Language Model Generation via Continuous Latent Sentence Spaces.

---

### Writing:

- The paper is not well written in its current form. While it describes what the authors did, the presentation reads more like a project report than a scientific paper suitable for conference submission. The writing lacks a clear narrative, logical flow, and proper academic tone. I recommend that the authors thoroughly revise the paper to improve its organization, clarity, and coherence.

- Sections 1 and 2 currently lack citations. The authors should incorporate relevant references to prior work to properly situate their study within the broader context of existing literature.

**Questions:**

- Figure 1: Why is a bidirectional decoder used instead of a standard decoder? During training, the task is reconstruction, but it is unclear whether a bidirectional decoder provides any advantage during inference. Could the authors clarify this design choice?

- Equations (4) and others: It would be clearer to use $\mathbf{x}$ to represent a complete sentence in the loss functions, rather than the scalar $x$.

---

### Official Review · Reviewer_3Yuy · 2025-10-31

**Soundness:** 2
**Presentation:** 2
**Contribution:** 1
**Rating:** 2
**Confidence:** 4

**Summary:**

This paper introduces a VAE-based language model named ​VALM (Variational Autoencoder Language Model)​, designed to address the ​high sequential decoding issue​ inherent in traditional Autoregressive Language Models (ARLMs) during inference, which arises from their token-by-token generation process. The core idea of VALM is to apply the ​Variational Autoencoder (VAE)​​ generative paradigm to discrete text, enabling ​parallel, single-pass text generation.

Specifically, VALM employs a bidirectional Transformer encoder to compress an input sequence into a ​global latent variable $z$. A bidirectional Transformer decoder then uses this single variable to predict logits for all token positions in parallel within a single forward pass.This design also prevents the common "posterior collapse" issue in text VAEs.

Paper developed a prototype, ​VALM-1, which demonstrates the ability to generate coherent 32-token spans in a single pass. Preliminary scaling law studies on datasets like TinyStories and WikiText-103 indicate that the model's performance (ELBO loss) improves predictably with increases in both model parameters and data volume, all without exhibiting KL collapse.

**Strengths:**

1. The instruction and method part are easy to follow.
2. The experiment clearly shows that VALM can maintain single-pass coherence within a short span.

**Weaknesses:**

1. The model's architecture is very traditional and lacks innovation.
2. VALM-1 can only generate fixed-length text of 32 tokens, and its results may not generalize to larger vocabularies, longer contexts, or more diverse domains.
3. The paper does not systematically compare VALM with autoregressive models of comparable scale, lacking performance benchmarks under the same dataset and evaluation metrics.
4. The experiments shown in the paper are very preliminary, lacking evidences to show this paradigm is promising on some  on some real-world applications.

**Questions:**

1. The model relies entirely on the latent variable $z$ to establish dependencies between all tokens. How does this design impact its performance in modeling long-range dependencies, and does it imply a fundamental ceiling on the effectiveness of VAEs for language modeling?
2. The paper primarily relies on qualitative examples to demonstrate coherence, lacking quantitative comparisons against baseline models on standard metrics. Does this omission suggest that the VALM framework faces challenges in being fairly evaluated within established benchmarking paradigms?
3. Table 3 indicates a relatively weak scaling exponent with respect to token count for WikiText-103 (0.09). Could this suggest that the VALM architecture has inherent limitations in leveraging increased data to improve performance on more complex datasets?
4. Table 2 and the main text highlight specific errors and limitations in the generated text. What is the hypothesized root cause of these errors? Would increasing the dimensionality of the latent variable $z$ be a viable strategy to mitigate these issues?

---

### Official Review · Reviewer_qsMG · 2025-10-31

**Soundness:** 2
**Presentation:** 2
**Contribution:** 3
**Rating:** 2
**Confidence:** 3

**Summary:**

The paper proposes VALM, a variational autoencoder language model that generates text non-autoregressively in a single forward pass using a global latent vector. This design aims to overcome the sequential bottleneck of autoregressive models and demonstrates a prototype implementation (VALM-1) capable of producing short, coherent text sequences.

**Strengths:**

- The paper introduces a novel and conceptually interesting approach, a non-autoregressive VAE for text generation, that challenges the dominant autoregressive paradigm.
- The prototype implementation demonstrates the feasibility of one-pass text generation and shows that variational methods can be applied effectively to discrete sequences.

**Weaknesses:**

- The evaluation is very limited, relying on small-scale qualitative examples (e.g., short passages from bAbI and TinyStories) without quantitative or comparative benchmarks.
- The work appears to be at an early, exploratory stage, with limited empirical validation and scope. As such, it might be more appropriate as a position or concept paper rather than a full research contribution.

**Questions:**

See my comments above.

---

### Note · Authors · 2025-11-21

**Comment:**

We are withdrawing this submission due to insufficient experimental validation and comparison to prior methods. We sincerely thank the reviewers for their time and feedback.

**Withdrawal Confirmation:**

I have read and agree with the venue's withdrawal policy on behalf of myself and my co-authors.